# Chess Practice as a Protective Factor in Dementia

**DOI:** 10.3390/ijerph16122116

**Published:** 2019-06-14

**Authors:** Manuel Lillo-Crespo, Mar Forner-Ruiz, Jorge Riquelme-Galindo, Daniel Ruiz-Fernández, Sofía García-Sanjuan

**Affiliations:** 1Department of Nursing, Faculty of Health Sciences, University of Alicante, San Vicente del Raspeig, 03690 Alicante, Spain; manuel.lillo@grupohla.com (M.L.-C.); jrgrqlm@gmail.com (J.R.-G.); sofia.garcia@ua.es (S.G.-S.); 2Department of Computer Sciences, Advanced Polytechnic School of Alicante, University of Alicante, San Vicente del Raspeig, 03690 Alicante, Spain; druiz@ua.es

**Keywords:** dementia, Alzheimer disease, chess, protective factor, scoping review

## Abstract

Background: dementia is one of the main causes of disability and dependency among the older population worldwide, producing physical, psychological, social and economic impact in those affected, caregivers, families and societies. However, little is known about dementia protective factors and their potential benefits against disease decline in the diagnosed population. Cognitive stimulating activities seem to be protective factors against dementia, though there is paucity in the scientific evidence confirming this, with most publications focusing on prevention in non-diagnosed people. A scoping review was conducted to explore whether chess practice could mitigate signs, deliver benefits, or improve cognitive capacities of individuals diagnosed with dementia through the available literature, and therefore act as a protective factor. Methods: twenty-one articles were selected after applying inclusion and exclusion criteria. Results: the overall findings stress that chess could lead to prevention in non-diagnosed populations, while little has been shown with respect to individuals already diagnosed. However, some authors suggest its capacity as a protective factor due to its benefits, and the evidence related to the cognitive functions associated with the game. Conclusion: although chess is indirectly assumed to be a protective factor due to its cognitive benefits, more studies are required to demonstrate, with strong evidence, whether chess could be a protective factor against dementia within the diagnosed population.

## 1. Introduction

According to a report published by the World Health Organization (WHO) in 2015, the number of people aged above 60 years will double between 2000 and 2050 [1]. In line with this, the worldwide Alzheimer’s Associations have stated that dementia, concretely Alzheimer disease (AD), is suffered by 47.5 million people in the world, and every year, 7.7 million new cases are declared. It is one of the main causes of disability and dependency among the older population due to the physical, psychological, social, and economic impact on caregivers, families, and societies. Moreover, dementia is an AD salient clinical feature, and AD is the first cause of dementia [1], representing approximately 70% of all dementia cases. Consequently, the older populations in the world have the highest probability of experiencing disorders in cognitive capabilities such as neurodegenerative diseases. For that reason, and due to the lack of early onset dementia detection strategies worldwide, it is crucial nowadays to identify factors that can positively influence the cognitive impairment process mostly experienced by the population diagnosed with dementia.

At this point it is important to discuss about the meaning of some terms used in published literature regarding our research topic such as “prevention”, “early diagnosis”, and “protective factor”. These concepts are sometimes interpreted equally or as synonyms, and consequently generating confusion. All these terms are related, and though they refer to different things, they point out different stages of the clinical process in which diagnosis marks the principle milestone. To clarify this, we have used MeSH definition terms as a starting framework:-**Prevention**: “Disease prevention is a procedure through which individuals, particularly those with risk factors for a disease, are treated in order to prevent a disease from occurring” [2]. In this case, the disease has not started or at least has not been detected yet.-**Early diagnosis**: “Methods to determine in patients the nature of a disease or disorder at its early stage of progression. Generally, early diagnosis improves prognosis and treatment outcome” [3].-**Protective factor**: “An aspect of personal behavior or lifestyle, environmental exposure, or inborn or inherited characteristic, which, on the basis of epidemiologic evidence, is known to be associated with prevention or mitigation of a health-related condition considered important to prevent” [4]. This concept has a narrow relation with the previous ones and, therefore, we are assuming that a protective factor is an aspect that could appear either linked with prevention or even when the diagnosis has been established (enhancing the mitigation of such condition) and also in early detection.

In line with the previous statements, the research question that emerged was: is chess a protective factor, and what are its benefits for people diagnosed with Dementia?

Signs of Dementia include memory loss that disrupts daily life, problem-solving challenges, and time/place confusion. Brains of Alzheimer’s patients have plaque and tangles along with a protein build-up between nerve cells and protein build up inside nerve cells. Although plaque and tangles develop normally as people age, patients with AD have more than average.

Researchers have found a link between playing chess and reducing the possibility of developing dementia, AD, and other debilitating mental illnesses. The *New England Journal of Medicine*, in the article published by Coyle [5], reported that individuals older than 75, engaged in leisure activities including chess, were apt to delay developing signs of dementia when compared to people who did not play. This study, which lasted five years and included almost 500 participants, indicates that involvement in at least ten mind-exercising activities per group, delays early onset of AD by nearly 1.5 years. Individuals who played board games were over 35 percent less likely to develop dementia than those who participated in board games only occasionally or even rarely. In fact, people over the age of 75 that partake in leisure activities such as chess that stimulate the brain were less likely to develop signs of dementia.

Chess seems like a treatment that works to prevent or protect before disease onset, though little is known about its benefits after the disease has been identified and diagnosed. Research shows that chess affects specific areas of the brain, and this stimulation shifts with the problems that a chess player faces during the game. What we know until now is that chess game lends itself to a variety of complexities from various patterns to complex calculations that stimulate players’ brains [5].

Paying attention to the cultural scope of our research topic, we identified paucity in the evidence published about research conducted in those countries where chess is practiced traditionally, even detecting the lack of basic epidemiological investigations about dementia. According to Kiejna et al. [6], there is a strong need for epidemiological studies in Eastern and Middle Europe, as well as for greater coordination and standardization of methods to improve the quality and comparability of epidemiological data to determine the prevalence rates of dementia in all EU countries. They were able to find few regional and country-specific epidemiological studies of various kinds (population-based, cohort, cross-sectional studies), conducted on different restricted population groups of patients (from neurological units, out-patients units, residential homes). No studies were identified from most of the countries taken under consideration, and the ones we found were characterized by an immense diversity with a considerable degree of clinical and methodological variations. In addition, the few studies published suggest prevalence rates of dementia in Eastern Europe similar to those in Western Europe. Moreover in the case of some Asiatic countries, the GO game—a sort of chess game played in China, Japan, and Korea—has been shown to exhibit positive results for diagnosed populations.

The concept of cognitive reserve (CR) explains individual differences in susceptibility to age-related brain changes and pathological changes, such as those occurring in Alzheimer’s disease. Individuals with higher CR can tolerate more pathologies, so the point at which cognitive functions begin to be affected will manifest later than in those with lower cognitive reserve. In other words, there is a reduced risk of developing Alzheimer’s disease in individuals with a higher level of education and, therefore, they will also be able to maintain their activities of daily living for longer [7].

### Aim

According to literature published to date, since leisure activities including chess, delay the development of dementia signs, it is considered a protective factor against dementia not only in prevention but also when the disease has been diagnosed, regardless of the stage. In addition, using the available literature, we explore whether chess practice can produce benefits or improve cognitive capacities of people with dementia once the syndrome is clinically identified, and whether it acts as a dementia protective factor at any time during the ageing process.

## 2. Materials and Methods

### 2.1. Design

The study design corresponds to the so-called scoping review, a type of review study that aims to identify relevant studies on a topic (in this case, explore whether chess practice can produce benefits or improve cognitive capacities of dementia patients or not through the available literature and therefore act as a protective factor) [8].

In contrast to the traditional systematic review, which aims to answer a very specific question with a population, intervention, comparison, and results (PICO) format, the scoping review answers a broader and more flexible question, characterized by elements of the population, concept, and context (PCC) format, the design choice is left to the researchers, allowing them to make a description in relation to time (when it was published), location (country), source (reviewed by experts or grey literature), and/or origin (of health or academic discipline). This results in an overview of emerging fields that, due to the diversity of articles and methods, make it difficult to ascertain the existing body of knowledge [9].

Towards formulating a proper scientific query, the PPC scheme was followed. Thus, the question “May chess help improve cognitive capacities of elder people diagnosed with dementia/AD or at least delay its onset?” was used as a starting point.

### 2.2. Literature Search

A literature review among scientific literature in English and Spanish language databases including Cuiden, Cinahl, Lilacs, Medline (through PubMed), and Scopus, was conducted. All the papers regarding the target topic were selected. Moreover, in order to explore all the sources of information, we searched in the referral section of the different studies included, and using search engines such as Google. Finally, chess game blogs were also reviewed, as well as the Enterrian Chess Federation (FEDA) webpage, to consult expert opinions on the topic, as well as National Statistics Institute of Spain (INE) and WHO webpages, to gather statistical data on the topic.

The keywords used for the study in English and Spanish were: dementia, Alzheimer disease, chess, protective factor, scoping review. Furthermore, the Boolean operators used were: AND and OR. All this information is outlined in Table 1.

### 2.3. Inclusion and Exclusion Criteria

Studies meeting the following criteria were included: scientific journal papers, systematic reviews with or without meta-analysis, experimental studies, descriptive studies, and bibliographic reviews published in English or Spanish languages.

All papers related to animals and thus, restricting the query to human beings exclusively, were excluded.

Finally, no search restrictions were imposed owing to the scarceness of available bibliography on the topic.

### 2.4. Data Extraction

A final amount of 413 papers were identified. Inclusion and exclusion criteria were applied, by the authors, to the titles and abstracts identified through the literature review, in order to determine their relevance.

In a first review, 392 were discarded due to lack of compliance with the inclusion criteria. At this point, decisions were monitored by the research team.

As a result of this process, 21 studies were finally included for further in-depth review. The search outcome is shown in Table 1.

A protocol was designed to present data in a comprehensive classification including characteristics of the studies selected such as author, year of publication, study objective(s), design and population, social context and sociodemographic item (sex, age, country where the study was conducted), and main results (Table 2).

## 3. Results

Effortful mental activity produces and strengthens synaptic connections and stimulates neurogenesis process. Thus, it promotes plastic changes in the brain that slow down the symptoms of dementia. In agreement with work by Coyle [5], our results highlight that seniors should be encouraged to read, play board games like chess, and dance, for example, because these types of activities or leisure activities, as referred by the literature, when carried out at the beginning of the disease or syndrome, enhance their quality of life and fortify their cognitive functions. The results of this scoping review are presented through themes that came up during analysis and reflect chess’s cognitive stimuli. We analyzed those studies by firstly classifying them according to the chess game’s cognitive functions they were referring to, then paying attention to the benefits of those functions and finally focusing on the evidences reported regarding diagnosed population:

### 3.1. High Mental Activities such as Chess to Prevent Dementia

Most studies highlight benefits regarding prevention instead of focusing on the diagnosed population. Some authors such as Scarmeas et al. [10] and Dowd et al. [12] propose that engagement in leisure activities may reduce the risk of incident dementia, providing a reserve that delays the onset of clinical manifestations of the disease. Different follow-up studies suggest a relationship between the degree of leisure activity such as chess and the risk of developing dementia or AD [11]. The results show that reading, meeting friends, or doing pleasant activities were the most strongly associated with a reduced risk of dementia [10], and so chess could be included in the same list. Furthermore, people with a high mental activity levels have a 33% lower risk of developing AD [12].

A five-year study with 488 participants showed that involvement in at least 11 mind exercising activities per week versus a control group that engaged in 4 or less activities per week, delayed by 1.3 years the Dementia onset [13].

Some studies show that older people who frequently engage in mentally stimulating activities have fewer possibilities to develop AD or experience cognitive decline [14]. Being cognitively active, might contribute to fortifying cognitive reserve and enhancing adaptation to age-related pathologic changes [15].

Dartigues et al. [16] observed in their study that board game players have a 15% lower risk of developing dementia (95% CI 0.74 to 0.99; *p* = 0.04). After a 20 years follow-up, 830 cases of early stage dementia (27.8%) were observed. Though the risk of dementia was significantly reduced in board game players (*p* < 0.001) after a three-year follow-up, 3% of board players developed dementia versus 6% of non-players, 16% versus 27% after ten years and 47% versus 58% after twenty years.

### 3.2. Chess to Postpone the Development of Dementia

Moreover, when talking about neurodegenerative diseases and early detection, some studies focus on diagnosed populations while others concentrate on early onset. The pathological features of AD are most profound in the limbic system and temporal, frontal, and association neocortices, and basal forebrain areas involved in learning, memory, emotion, judgment, abstraction, language, and executive functions [17]. It has been therefore hypothesized that intellectual activities involving learning and memory would be most protective against the development of the disease. Among these activities, chess stands out as an activity gathering all these features [18]. Short-term and long-term memory usage, as well as calculation and visual–spatial abilities and critical thinking are five cognitive areas, with are all covered by this easy to learn and practice mind sport.

Archer et al. [19] studied the case of a chess player and highlighted the difficulties encountered in assessing patients with superior premorbid function in the early stages of Alzheimer’s disease, and reveals the value of serial Magnetic Resonance Imaging (MRI) and neuropsychological assessment in detecting and monitoring early neurodegenerative diseases. Li et al. [20] showed that the prevalence rate of dementia increased with age in Gushan, and concluded that factors as playing poker/chess more frequently, as well as taking good care of families tend to help reduce or postpone the development of dementia.

A study by Lin et al. [21] confirms that “GO game” (a kind of Chinese chess game over 5000 years old) can improve quality of AD patients. In fact, this kind of game involves the changes associated with many cognitive functions, including learning, abstract reasoning, and self-control, which facilitate cognitive behavioral therapy. Another study [22], demonstrated that brain training needed for playing this kind of Chinese game causes structural changes that are particularly helpful, in terms of engaging in such foundational tasks as learning, abstract reasoning, problem solving and self-control.

### 3.3. Cultivating Intellectual Activities since Youth Seems to Protect against Dementia

There is evidence that AD and good brain health have multifactorial causes like genetics or individual differences in cognitive reserve; quality of sleep, anxiety or mood disorders, physical exercise or a healthy diet are key factors in the AD approach [23]. However, it cannot be left out that cognitive stimulation improves mental health and protects against depression symptoms, which are very important for AD patients’ life quality [24].

Therefore, some studies [25] show that participating in intellectual activities during youth may protect against AD after years. However, there is not a lot of evidence on what can be done when AD appears in later stages of life [26]. Older people use different brain areas compared to young people, even when they perform the same activity [25]. This phenomenon allows the remaining of cognitive functions during life.

Some studies reflect that cognitive activities reduce the amyloid build up in the brain [27]. There is a hypothesis that amyloid build up probably starts 7–10 years before AD symptoms appear. For that reason, there could be a chance that healthcare professionals could attempt to stop disease progression, preventing symptoms of AD such as memory loss.

In addition to these factors, the ageing process depends on physical, cognitive, and mental factors. A balanced diet, practicing physical exercise and controlling any eventual diseases during this stage, are items that can completely change the quality of life of a person. As for cognitive factors, previous learning and educational level will be capital determining a predisposition to carry out these healthy behaviors [23]. Some studies show that mental activity brings several benefits to an individual’s health [24] that could be considered as improvements in cognitive capabilities. It has been demonstrated that mind exercises, physical activity, and board games stimulate all six cognitive areas of the brain at the same time being a useful tool to reduce the impact of ageing on cognitive functions. However, those studies do not support the idea of chess as a protective factor though stress the idea of chess as preventive one with positive benefits during the ageing process.

### 3.4. Leisure Activities such as Chess or Similar Ones and Cognitive Decline

There is another hypothesis that some epidemiologic studies reported, stating the association between high rates of leisure activities in old age and a lower cognitive decline, based on the cognitive reserve theory [28]. People with a higher cognitive reserve have more efficient neuronal networks, or they have the possibility to use alternative networks, which delays AD incidence [29].

Further analysis demonstrated those that played only games reduced their risk by 75% and those that played musical instruments reduced theirs by 64%. Crossword puzzle enthusiasts get a 38% lowered risk [5].

Finally, Fritsch et al. [30], in their study concluded that participation in leisure activities to improve cognitive stimulation is associated with reduced odds of having AD. People who participated in many types of mental activities reduced their chances to develop AD in comparison with those who only did one type.

## 4. Discussion

The available literature shows that there is evidence supporting the fact that intense mental activities are preventive for dementia and AD and therefore its benefits are obvious. Most papers we reviewed supported that leisure activities including chess could have a protective effect for dementia and AD development. However, even though publications suggest the protective condition of chess game for dementia in terms of prevention, this does not justify its practice as a protective factor. Moreover, all the studies showed that cognitive stimulation with leisure activities has many benefits for Alzheimer’s disease patients’ cognitive health.

Dowd et al. [12,31] conclude in their review that chess may be helpful in preventing AD, since it has been proved that is a skill-based game rather than depending on general intelligence; but not only this, chess could also have a relevant role in AD therapy, once the disease has been established, during its early stages though this is just the authors’ suggestion not based on clear evidence. This duality in chess (and other mind games) usefulness against dementia and AD raises its importance even more, and justifies the convenience of having it implemented in standardized nursing intervention programs to approach this condition [32].

Lin et al. [21] performed their clinical trial based on the GO game, a board game with features like those of chess, with simple rules and rich strategies, but emerging from Chinese culture. They point out that this game, which is very popular in China, Japan, and Korea, with the additional benefit that it is especially attractive among the elderly adults. The fact of this activity being popularized and widely spread among this kind of population make it highly suitable to be implemented and its practice potentiated in AD patients. In our environment, chess cannot be considered to be a game commonly played by the elderly, but conducting efforts in that sense may prove useful, if not on a population-level, at least in elderly care centers or leisure gathering spots. It also must be stressed that an activity being genuinely attractive and joyful, brings an additional positive effect in protecting against symptoms such alexithymia (defects in regulating feelings), which typically co-occurs with depression [33,34]. Lee et al. [22] further support these effects through their imaging study that displays the neuroplastic changes associated with long-term playing the GO game. It was evidenced that larger regions of white matter were developed in areas related to attention control, working memory, executive regulation and problem solving, setting an initial point of these features being helpful for improving capacities that can facilitate education and cognitive therapies oriented to approach dementia.

Although most scientific evidence tends to support the idea of mental exercises as AD protective factors, some papers against this idea can be found in the literature, as well as others that just treat it as a recommendation. False hope and apparent blaming dementia patients for their condition due to having failed to exercise their brains sufficiently are the main arguments suggesting that mind exercising could not be only positive but also not harmless in this population [23]. In this sense, some randomized controlled trials of cognitive training with older adults, such as ACTIVE study (Advanced Cognitive Training for Independent and Vital Elderly), where patients were assigned to wither no training, or receiving it in memory, reasoning or speed of processing, with their effects being followed-up for 2 years, showed that cognitive training programs had no significant effect on measures of everyday functioning, and remaining unknown whether eventual rates of AD would be reduced [35].

As part of the limitations of this scoping review, no studies that reflect the benefits of chess in the different dementia types (AD, dementia with Lewy bodies, frontotemporal dementia, vascular dementia, and mixed dementia) were found. In addition, we did not find publications comparing the benefits of playing chess throughout life and playing chess just at an older age. Finally, some of the studies we found did not define independent variables (sex, age, level of studies, profession, culture, etc.) of the dependent ones (playing chess and developing dementia).

## 5. Conclusions

Despite all the benefits listed throughout this review, and despite the data existing against their positive effects being scarce, the evidence is currently not strong enough to infer a direct causal relationship or recommend one leisure activity over the other [36]. Thus, chess could be considered a protective factor against dementia and cognitive decline in older people, particularly due to the enhancement of cognitive reserve. However, skill-based activities rather than those mainly depending on general intelligence seem to be more convenient, since its implementation appears to be not as challenging in a population with special characteristics such patients with dementia, where skill learning is still a possibility [12].

Given this perspective and current evidence, there is no doubt that more controlled trials are needed to assess the protective effect of cognitive activities on the risk of dementia [37], though even more for diagnosed populations. These trials must be run over significantly long times and assess different variables, to get a view of the effect of these factors in protecting dementia in the long run. This effortful activity is recommended due to the incapacity of detecting the disease in the early stages. Even the most recent evidence suggests that the disease manifests around 18 years after the onset [38]. This may be the way to provide mental activities and exercising with their proper part in AD and dementia approach and to set the starting point for their systematic implementation in prevention and treatment.

Considering our results, the practice of chess is a protective strategy in the development of dementia from a preventive perspective, and even though the evidence is weak to demonstrate its role as a protective factor, other evidence indirectly related to chess and based on its cognitive stimulating functions suggest it may work as a protective factor.

## Figures and Tables

**Table 1 ijerph-16-02116-t001:** Flowchart: search strategies in on-line resources.

Database	Search Equation	Articles Found	Articles Selected
**PubMed**	Dementia **AND** chess	6	4
	Alzheimer disease **OR** dementia **AND** chess **OR** chess game **AND** protective factor	2	2
	Chess **AND** dementia **OR** Alzheimer’s disease **AND** memory **AND** perception **AND** attention	216	4
	Chess **OR** chess game **AND** dementia **OR** Alzheimer’s disease	123	4
**Scopus**	Dementia **AND** chess	5	3
	Dementia **AND** chess **AND** Alzheimer’s disease	2	2
	Dementia **OR** Alzheimer’s disease **AND** chess **OR** chess game	4	0
	Chess **AND** dementia **OR** Alzheimer’s disease **AND** memory **AND** perception **AND** attention	0	0
	Chess **AND** dementia **OR** Alzheimer’s disease	15	2
**CINALH**	Chess **AND** dementia **AND** Alzheimer’s disease	28	0
**Google Academic**	Chess **AND** dementia	0	0
**Google**	Chess **AND** dementia	4	0
**LILACS**	Chess **AND** dementia	8	0
**CUIDEN**	Ajedrez **AND** demencia	0	0

**Table 2 ijerph-16-02116-t002:** Summary of the studies included.

ARTICLE	DESIGN	SAMPLE	OBJECTIVES	RESULTS
[10]	Cohort study.	1772 old people. New York.	To determinate whether leisure activities modify the risk for the incident of dementia.	The risk of dementia decreased in subjects with high leisure activities (RR: 0.62; 95% CI: 0.46 to 0.83).
[11]	Cohort study.	2040 old people. France.	To study the relationship between social and leisure activities, and risk of subsequent dementia in older community residents.	All but one of the social and leisure activities noted were significantly associated with a lower risk of dementia. Travelling (RR: 0.48, 95% CI: 0.24–0.94), odd Jobs or knitting (RR: 0.46, 95% CI: 0.26–0.85), and gardening (RR: 0.53, 95% CI: 0.28–0.99) remained significant.
[12]	Literature review.	-	To review studies on AD and mental activity to see whether the second can be helpful in preventing the first.	It appears that engaging the elderly in activities that use the brain can be helpful in stemming the epidemic that is AD.
[13]	Cohort study.	1348 participants. New York.	To examine trends in dementia incidence and concomitant trends in cardiovascular comorbidities among individuals aged 70 years or older, who were enrolled in the Einstein Aging Study between 1993 and 2015.	150 incidents of dementia cases developed during 5932 person-years. Dementia incidence decreased in successive birth cohorts. Incidence per 100 person-years was 5.09 in birth cohorts before 1920, 3.11 in the 1920 through 1924 birth cohorts, 1.73 in the 1925 through 1929 birth cohorts, and 0.23 in cohorts born after 1929.
[14]	Longitudinal cohort study.	700 old people.	To examine the relationship between cognitive activity and development of AD.	More frequent participation in cognitive activity was associated with reduced incidence of AD (HD: 0.58; 95% CI: 0.44, 0.77).
[15]	Perspective.	-	To describe cognitive stimulation therapy development and evaluation, its use in clinical setting and issues for future investigation.	-
[16]	Prospective population-based study.	3675 old people. France.	To study the relationship between board game playing and risk of subsequent dementia in the Paquid cohort.	The risk of dementia was 15% lower in board game players compared to non-players (HR: 0.85, 95% CI: 0.74 to 0.99; *p* = 0.04).
[17]	Review.	1 case 100 studies	To review and attempt to integrate the disparate elements of the disease into a coherent whole, perhaps helping to focus future investigative efforts on developing rational treatments.	An integrated diagnostic and therapeutic approach to this complex and tragic disorder may still seem remote, the current rate of scientific progress indicates that some level of practical success may come sooner than one might think.
[18]	Case-control study	193 case people and 358 control people	To evaluated the relationships between non-occupational activities and AD in a case-control study.	The odds ratio for AD in those performing less than the mean value of activities was 3.85 (95% CI: 2.65–5.58, *p* < 0.001).
[19]	N = 1 study	1 case	Report a case of a chess player who presents cognitive disorders.	Cognitive function appeared and AD was diagnosticated.
[20]	Parallel group.	29 participants.	Report dates to provide research with new materials to further explore the human brain.	-
[21]	Case-control study.	147 old people. China.	To explore the functions of GO game in AD patients.	GO game intervention ameliorates AD manifestations.
[22]	Case-control study.	16 experienced Baduk players. Korea.	To elucidate the brain’s structural development as it relates to the cognitive components needed to play Baduk, based on White matter neuroplastic changes.	Long-term trained Baduk players developed larger regions of white matter with increased fractional anisotropy values in different areas that are related to attention control, working memory, executive regulation, and problem solving.
[23]	Literature review.	-	To determine what literature states about mental practice as a protective factor against dementia and AD.	There is no convincing evidence of memory practice and other cognitively stimulating activities be sufficient to prevent AD.
[24]	Longitudinal cohort study.	801 old people. USA.	To test the hypothesis that frequent participation in cognitive activities is associated with a reduced risk of AD.	During an average of 4.5 years of follow-up, 111 persons developed AD. In a proportional hazards model that controlled for age, sex, and education, a 1-point increase in cognitive activity score was associated with a 33% reduction in risk of AD (HR: 0.67; 95% CI: 0.49–0.92)
[25]	Opinion article.	-	-	-
[26]	Systematic review.	55 studies.	To present evidence from observational studies concerning the impact of leisure activities on the risk of dementia and cognitive decline and provide evidence from intervention studies on the topic.	A protective effect of mental activity on cognitive function has been consistently reported in both observational and interventional studies. The association of mental activity with the risk of dementia was robust in observational studies but inconsistent in clinical trials. Current evidence concerning the beneficial effect of other types of leisure activities on the risk of dementia is still limited and results are inconsistent.
[27]	Divulgative study review.	-	-	-
[28]	Cohort study-	5.698 old people. France.	To examine the association between leisure activities and risk of incident dementia and its subtypes.	Stimulating leisure activities were found to be significantly associated with a reduced risk of dementia (HR: 0.49, 95% CI: 0.31; 0.79) and AD (HR: 0.39, 95% CI: 0.21; 0.721).
[29]	Cohort study.	283 old people. New York.	To determine whether pre-diagnosis leisure activities modify the rate of cognitive decline in patients with AD.	Each leisure activity was associated with an additional yearly decline of 0.005 of a z-score unit in cognitive score (*p* = 0.17).
[30]	Case-control Study.	264 cases and 545 controls (809 participants). Ohio.	To study the association between participation in different types of mentally stimulating leisure activities and status as AD case or normal control.	Logistic regression analysis indicating that adjusting for control variables, greater participation in novelty-seeking, and exchange-of-ideas activities were significantly associated with decreased odds of AD.

Abbreviations: -AD: Alzheimer Disease; -CI: Confiance Interval; -RR: Relative Risk; -HR: Hazard Ratio; -HD: Hazard Deviation.

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
