# Peer review of "Chess Practice as a Protective Factor in Dementia"

_ijerph, 2019, doi:10.3390/ijerph16122116_

Round 1

Reviewer 1 Report

The article focuses on important topic - potential prevention of dementia syndrome, but the design of the paper and lay out of ideas and results are confusing. The concept of prevention disease (or syndrome) itself and delay of the symptoms are interchangeably used and mixed in the text. In lines 65-66 is written: "Chess seems like a treatment that works to prevent though less is known about its positive benefits when the disease is on." But the next line contradicts since speaks "In fact, people over the age of 75 that partake in leisure activities that  stimulate the brain were less likely to develop signs of dementia." 

The paragraph (line 71-80) is out of scope of the article since speaks about epidemiological studies and in the manner which is not related to the title of the article.

The aim itself seems to be contradictory (lines 87-88), since authors intend to explore effect of chess on persons with dementia and at the same time - on prevention ("To explore whether chess practice can produce benefits or improve cognitive capacities of dementia patients or not through the available literature and therefore act as a protective factor." The same is in lines 103-105.

Line 114 - it is written that one of the keywords was "protective factor", but in the table below this keyword is not included.

The results not adequately enough correspond to the title of the article: section "3.1. High mental activities to reduce dementia" has no information about chess at all. The same is with the section "3.3. Cultivating intellectual activities since youth protects against dementia" and "3.4. Leisure activities & dementia". In the results section there is no clear distinction between effect of chess on prevention and effect on patients with already established dementia (according the aim).

The discussion part focuses just on few studies.

Author Response

Reviewer 1

The article focuses on important topic - potential prevention of dementia syndrome, but the design of the paper and lay out of ideas and results are confusing. The concept of prevention disease (or syndrome) itself and delay of the symptoms are interchangeably used and mixed in the text. In lines 65-66 is written: "Chess seems like a treatment that works to prevent though less is known about its positive benefits when the disease is on." But the next line contradicts since speaks "In fact, people over the age of 75 that partake in leisure activities that stimulate the brain were less likely to develop signs of dementia." 

We appreciate your input. With them, we have carried out a modification of said section:

At this point it is important to discuss about the meaning of some terms used in published literature regarding our research topic such as “prevention”, “early diagnosis” and “protective factor”. These concepts are sometimes interpreted equally or as synonyms and consequently generating confusion. All of them are one-another related though they refer to different things and what is more important, they point out different stages of the clinical process in which the diagnosis marks the principle milestone. To clarify this, we have used Mesh definition terms as a starting framework:

- Prevention: “Disease prevention is a procedure through which individuals, particularly those with risk factors for a disease, are treated in order to prevent a disease from occurring”[2].In this case, the disease has not started or at least has not been detected yet. 

- Early diagnosisMethods to determine in patients the nature of a disease or disorder at its early stage of progression. Generally, early diagnosis improves prognosis and treatment outcome”[4].

- Protective factorAn aspect of personal behavior or lifestyle, environmental exposure, or inborn or inherited characteristic, which, on the basis of epidemiologic evidence, is known to be associated with prevention or mitigation of a health-related condition considered important to prevent”[3]This concept has a narrow relation with the previous ones and therefore we are assuming that a protective factor is an aspect that could appear either linked with prevention or even when the diagnosis has been established (enhancing the mitigation of such condition) and also in early detection. 

In line with the previous statements, the research question that merged was: Is Chess Game a Protective Factor and which are its benefits for Dementia people?

The paragraph (line 71-80) is out of scope of the article since speaks about epidemiological studies and in the manner which is not related to the title of the article.

The aim itself seems to be contradictory (lines 87-88), since authors intend to explore effect of chess on persons with dementia and at the same time - on prevention ("To explore whether chess practice can produce benefits or improve cognitive capacities of dementia patients or not through the available literature and therefore act as a protective factor." The same is in lines 103-105.

We appreciate your input. With them, we have carried out a modification of said section:

Paying attention to the cultural scope of our research topic, we identified paucity in the evidences published about research conducted in those countries where chess is practiced traditionally even detecting the lack of basic epidemiological investigations about dementia.  According to Kiejna A et al. [6], there is strong need for epidemiological studies in Eastern and Middle Europe, as well as for greater coordination and standardization of methods to improve the quality and comparability of epidemiological data to determine the prevalence’s rates of dementia in all the EU countries. They were able to find few regional and country-specific epidemiological studies of various kinds (population-based, cohort, cross-sectional studies) and conducted on different restricted population groups of patients (from neurological units, out-patients units, residential homes). No studies were identified from most of the countries taken under consideration and the ones we found were characterized by an immense diversity with a considerable degree of clinical and methodological variations. In addition, the few studies published suggest prevalence rates of dementia in Eastern Europe similar to those in Western Europe. Moreover in the case of some Asiatic countries, the GO Game, a sort of Chess Game played in China, Japan and Korea, some studies have highlighted its positive results for diagnosed population.

Line 114 - it is written that one of the keywords was "protective factor", but in the table below this keyword is not included.

We appreciate your correction and we notify that it has already been modified in Figure 1 verifying that the results are correct.  

The results not adequately enough correspond to the title of the article: section "3.1. High mental activities to reduce dementia" has no information about chess at all. The same is with the section "3.3. Cultivating intellectual activities since youth protects against dementia" and "3.4. Leisure activities & dementia". In the results section there is no clear distinction between effect of chess on prevention and effect on patients with already established dementia (according the aim).

We have modified the titles based on their recommendations:

3.1. High mental activities such as Chess to prevent dementia.

3.3. Cultivating intellectual activities since youth seems to protect against dementia.

3.4. Leisure activities such as chess or similar ones & cognitive decline.

The discussion part focuses just on few studies.

We appreciate your input. With them, we have carried out a modification of said section:

32. Dowd, S.B. & Root, A. What chess has given us? Academic Erchange Quarter. 2003, 7, 1.

35. Li, L., Chang, H.J., Yeh, H., Hou, C.J.Y., Tsai, C.H., Tsai, J.P. Factors associated with leisure participation among the elderly living in longterm care facilities.Int J Gerontology. 2010, 4 (2), 69-74.

Reviewer 2 Report

International Journal of Environmental Research and Public Health-507818-review

Recommendation: Major revision

Reviewers' comments

Reviewer : General comment:

The topic of the present study is of high relevance for the readers of International Journal of Environmental Research and Public Health and definitely falls within the journal scope. The aim of the present study was to investigate whether chess practice can produce benefits or improve cognitive capacities of dementia patients or not through the available literature and therefore, act as a protective factor. The authors conducted a so-called scoping review. Twenty-one studies were found and included for further in-depth review. The results showed that chess, as other cognitive stimulating activities, may act as a potential tool to delay dementia onset and therefore inducing prevention. The authors conclude that more studies are warranted before any firm conclusions can be drawn.

I like the topic of the paper very much and overall the paper is well written. However, there are several major and minor revisions that the authors need to make. Depending on the quality of the revision, I am in favor (if they do a good revision job) or against (in case of a weak revision) publication.

Major revisions:

- Introduction (pages 1-2): Something went wrong with the numbering of the literature. Somehow your first reference ended up to be number four in the manuscript. Moreover, I cannot find the first three references in your reference list in the manuscript anymore. Please correct this.

- Throughout the manuscript: Moreover, something went wrong with the numbering of the manuscript pages as well. The first 6 pages are correct, but then it starts with page 1 again (e.g., page 1, page 2, page 3, page 4, page 5, page 6, page 1, page 2, page 3, page 4, page 5, page 6). Please correct this. I will in my comments from the Results section onwards continue with page 7, page 8, page 9, page 10, page 11, and page 12.

- Introduction (page 2, lines 81-82): The authors should explain a bit more into detail what the concept of “cognitive reserve” means, for instance, by giving a definition. Here, the authors could refer to the work by Stern (2002, 2012) in the Journal of the International Neuropsychological Society [https://www.ncbi.nlm.nih.gov/pubmed/11939702] or in the Lancet Neurology [https://www.ncbi.nlm.nih.gov/pmc/articles/PMC3507991/]. The point is that the readers should have a full understanding of the concept of cognitive reserve, in order to fully see the potential of your paper/the chess game as a protective factor.

- Introduction (pages 1-2): The authors give the study aim, but I miss their hypothesis/hypotheses at the end of the introduction. Please add. 1) I guess the authors expect/hypothesize that the chess game is a protective factor for dementia. 2) In addition, I expect the authors to hypothesize that the chess game is a prevention tool for dementia.

- Materials and Methods (page 3, lines 91-105): Why did the authors choose the “scoping review” procedure instead of the guidelines of the Preferred Reporting Items for Systematic Reviews and Meta-analysis (PRISMA)?

Moher, D.; Liberati, A.; Tetzlaff, J.; Altman, D.G.; The PRISMA Group. Preferred reporting items for systematic reviews and meta-analyses: The PRISMA statement. BMJ. 2009, 339, eb2535. [CrossRef] [PubMed]

I personally don’t get the point why the question “May chess help improve cognitive capacities of elder people diagnosed with dementia/AD or at least delay its onset?” could not be investigated with the systematic review or meta-analysis? In fact, the PRISMA guidelines are nowadays more or less the standard for conducting scientific reviews. Please explain.

- Materials and Methods (pages 3 and 4): The authors give a good description of their literature search and data extraction. However, I have a problem with the weighting of the studies. For instance, the conclusions drawn based on a case study (n=1) [REF 22] are not as strong as the conclusions drawn on a study (n = 5698) [REF 32]? Wouldn’t it be better to split the large data studies from the case studies? Please elaborate on this. It might be an option to tap this issue in the Discussion section of your paper.

- Results section of the paper (pages 7 and 8): I would like to see a clear result, showing how many of your 21 studies (in absolute numbers/percentages) found evidence in favor, mixed result, or evidence against a cognitive reserve effect of the chess game. I think that’s what the reader would like to see. In line with this comment, I would suggest adding a colon “Cognitive reserve” under which you write down “YES”, “PARTIAL”, “NO” so that the readers can immediately see whether the study found evidence in support or against the cognitive reserve effect of playing chess. Moreover, now you use the numbers “under ARTICLE”, but perhaps it better to mention the author + year of publication here as well. Instead “[13]” “Scarmeas et al. 2001 [13]”.

- Results section of the paper (pages 7 and 8): Could you give any information about how many years playing chess was found to postpone the development of dementia? For instance, lifelong bilingualism was found to delay the onset of dementia with 4 to 5 years.

- Discussion section of the paper (pages 8 and 9): One paragraph is needed discussing how these results should be seen in the light of the different types of dementia, e.g., Alzheimer dementia, dementia with Lewy bodies, frontotemporal dementias, vascular dementia, and mixed dementia? Are the cognitive reserve findings of the chess game the same for the different types of dementia or are they better in one specific kind of dementia?

- Discussion section of the paper (pages 8 and 9): Moreover, one paragraph is needed discussing the issue whether “playing chess your whole life” or “learning to play chess in older age” contributes to cognitive reserve (protection against dementia). This is an interesting issue if one considers “learning to play chess” as a clinical intervention technique in older adults.

- Discussion section of the paper (pages 8 and 9): The authors should discuss the methodological issue of “retrospective” versus “prospective” studies and why this is important in interpreting the results. Other methodological factors that could perfectly affect the study results are level of education, profession, culture, etc. I think these methodological affecting factors should be discussed more into detail. They are particularly important to correctly interpret the study results and are also very important with respect to future research.

- Discussion section of the paper (at the end, page 9, line 263): The authors should add a paragraph, discussing the limitations of the present study (see also my previous point 6).

- Discussion section of the paper (at the end, page 9, line 263): The authors should add a paragraph discussing what future research is needed based on the present study results, in order to make progress in the field. Please don’t forget to add the new references to your reference list.

- Conclusions (page 9, lines 279-284): Although I like your PhD project very much, this section should be deleted in the Conclusions section of the present paper.

- Reference list (pages 10-12): In general, the authors did a good job with respect to the Reference list. However, several references need to be corrected according to the reference style of International Journal of Environmental Research and Public Health.

Minor revisions:

- Somehow there is no subheading 2.3? You use “2.2.- Literature search” on page 3 and “2.4.- Data extraction” on page 4.

- On page 9, line 274 à “recomended” should be “recommended”. Please correct.

Author Response

Reviewer 2

General comment:

The topic of the present study is of high relevance for the readers of International Journal of Environmental Research and Public Health and definitely falls within the journal scope. The aim of the present study was to investigate whether chess practice can produce benefits or improve cognitive capacities of dementia patients or not through the available literature and therefore, act as a protective factor. The authors conducted a so-called scoping review. Twenty-one studies were found and included for further in-depth review. The results showed that chess, as other cognitive stimulating activities, may act as a potential tool to delay dementia onset and therefore inducing prevention. The authors conclude that more studies are warranted before any firm conclusions can be drawn.

I like the topic of the paper very much and overall the paper is well written. However, there are several major and minor revisions that the authors need to make. Depending on the quality of the revision, I am in favor (if they do a good revision job) or against (in case of a weak revision) publication.

Major revisions:

- Introduction (pages 1-2): Something went wrong with the numbering of the literature. Somehow your first reference ended up to be number four in the manuscript. Moreover, I cannot find the first three references in your reference list in the manuscript anymore. Please correct this.

Thanks for your appreciation. We have reviewed the numbering of the references and modified it.

- Throughout the manuscript: Moreover, something went wrong with the numbering of the manuscript pages as well. The first 6 pages are correct, but then it starts with page 1 again (e.g., page 1, page 2, page 3, page 4, page 5, page 6, page 1, page 2, page 3, page 4, page 5, page 6). Please correct this. I will in my comments from the Results section onwards continue with page 7, page 8, page 9, page 10, page 11, and page 12.

Many thanks for your corrections. We have reviewed the numbering of the pages and modified it.

- Introduction (page 2, lines 81-82): The authors should explain a bit more into detail what the concept of “cognitive reserve” means, for instance, by giving a definition. Here, the authors could refer to the work by Stern (2002, 2012) in the Journal of the International Neuropsychological Society [https://www.ncbi.nlm.nih.gov/pubmed/11939702] or in the Lancet Neurology [https://www.ncbi.nlm.nih.gov/pmc/articles/PMC3507991/]. The point is that the readers should have a full understanding of the concept of cognitive reserve, in order to fully see the potential of your paper/the chess game as a protective factor.

Thanks for your contributions. After reviewing this section we have made the following modifications:

The concept of cognitive reserve (CR) explains individual differences in susceptibility to age-related brain changes and pathological changes, such as those occurring in Alzheimer's disease. Individuals with higher CR can tolerate more pathologies, so the point at which cognitive functions begin to be affected will be later than in those with lower cognitive reserve. In other words, there is a reduced risk of developing Alzheimer's disease in individuals with a higher level of education and therefore they will also be able to maintain their activities of daily living for longer [7].

7. Stern, Y. Cognitive reserve in ageing Alzheimer’s disease. Lancet Neurol.2012, 11(11):1006-1012.

- Introduction (pages 1-2): The authors give the study aim, but I miss their hypothesis/hypotheses at the end of the introduction. Please add. 1) I guess the authors expect/hypothesize that the chess game is a protective factor for dementia. 2) In addition, I expect the authors to hypothesize that the chess game is a prevention tool for dementia.

Thanks for your contributions. After reviewing this section we have made the following modifications in “aim” and “abstract” sections: 

Aim:

According to literature published until now, since leisure activities including chess, delay the development of dementia signs, it is expected to behave as a protective factor of dementia not only in prevention but also when the disease has been diagnosed and whichever the stage is. In addition, we will explore whether chess practice may produce benefits or improve cognitive capacities of dementia people once the syndrome is clinically identified through the available literature and therefore act as a dementia protective factor during whichever time of the ageing process.

Abstract:

A scoping review was conducted to explore whether chess practice could mitigate signs, produce benefits or improve cognitive capacities of diagnosed dementia people through the available literature and therefore act as a protective factor.

- Materials and Methods (page 3, lines 91-105): Why did the authors choose the “scoping review” procedure instead of the guidelines of the Preferred Reporting Items for Systematic Reviews and Meta-analysis (PRISMA)?

Moher, D.; Liberati, A.; Tetzlaff, J.; Altman, D.G.; The PRISMA Group. Preferred reporting items for systematic reviews and meta-analyses: The PRISMA statement. BMJ. 2009, 339, eb2535. [CrossRef] [PubMed]

I personally don’t get the point why the question “May chess help improve cognitive capacities of elder people diagnosed with dementia/AD or at least delay its onset?” could not be investigated with the systematic review or meta-analysis? In fact, the PRISMA guidelines are nowadays more or less the standard for conducting scientific reviews. Please explain.

We agree with his assessment of the debate over whether a study can be considered a systematic review or a Scoping Review. In our case we consider it to be a Scoping Review because our research question is very broad, whereas if it were a systematic review it would have narrower parameters. The inclusion and exclusion criteria were not made from the beginning, as they should be in systematic reviews, but some were developed post hoc. Furthermore, the quality of the studies was not considered an initial priority.  We also chose scoping review, as it is a subject that is not clearly defined with parameters and there are gaps in the literature.

Armstrong R, Hall BJ, Doyle J, Waters E. ‘Scoping the scope’of a Cochrane review. J Public Health 2011, 33(1):147-150.

We also enclose a document from the PRISMA source to which you send us, where the parameters for evaluating a scoping review appear.

Tricco, A., Straus, S., & Moher, D. Preferred reporting items for systematic reviews and meta-analysis: extension for Scoping Reviews (PRISMA-ScR). Available online: http://www.equator-network.org/wp-content/uploads/2009/02/Executive-summary_ScR_Dec-9.pdf (accessed 20 May 2017).

- Materials and Methods (pages 3 and 4): The authors give a good description of their literature search and data extraction. However, I have a problem with the weighting of the studies. For instance, the conclusions drawn based on a case study (n=1) [REF 22] are not as strong as the conclusions drawn on a study (n = 5698) [REF 32]? Wouldn’t it be better to split the large data studies from the case studies? Please elaborate on this. It might be an option to tap this issue in the Discussion section of your paper.

I understand your assessment and we would agree with you if it were a meta-analysis type review, but in our case, the focus of the study was more on the results based on our topic to explore than the sample, that is, we did not give much weight in our review to the sample of the studies but to the objective of them and the obtained results.

We considerate that this article [before 22, now 19] is important in our study because it shows the limitations of currently available clinical investigation in the diagnosis of individuals with very early Alzheimer’s disease.

19. Archer, H.A., Schott, J.M., Barnes, J., Fox, N.C., Holton, J.L., Revesz, T., Cipolotti, L. Knight’s move thinking? Mild cognitive impairment in a chess player. Neurocase. 2006, 11(1):26-31.

- Results section of the paper (pages 7 and 8): I would like to see a clear result, showing how many of your 21 studies (in absolute numbers/percentages) found evidence in favor, mixed result, or evidence against a cognitive reserve effect of the chess game. I think that’s what the reader would like to see. In line with this comment, I would suggest adding a colon “Cognitive reserve” under which you write down “YES”, “PARTIAL”, “NO” so that the readers can immediately see whether the study found evidence in support or against the cognitive reserve effect of playing chess. Moreover, now you use the numbers “under ARTICLE”, but perhaps it better to mention the author + year of publication here as well. Instead “[13]” “Scarmeas et al. 2001 [13]”.

Thank you very much for your appreciation. Unfortunately, we can not classify the results of the different articles according to their cognitive reservation suggestion "YES", "PARTIAL" and "NO" since, as explained below, not all the articles expose the different independent variables (sex, age, level of studies, profession, culture, etc.) included in the studies and CR is defined as individual differences in susceptibility to brain changes related to age and pathological changes, such as those that occur in Alzheimer's disease . Individuals with higher CR can tolerate more pathologies, so the point at which cognitive functions begin to be affected will be later than those with less cognitive reserve. In other words, there is a reduced risk of developing Alzheimer's disease in individuals with a higher level of education and, therefore, they will also be able to maintain their activities of daily living for longer.

- Results section of the paper (pages 7 and 8): Could you give any information about how many years playing chess was found to postpone the development of dementia? For instance, lifelong bilingualism was found to delay the onset of dementia with 4 to 5 years.

- Discussion section of the paper (pages 8 and 9): Moreover, one paragraph is needed discussing the issue whether “playing chess your whole life” or “learning to play chess in older age” contributes to cognitive reserve (protection against dementia). This is an interesting issue if one considers “learning to play chess” as a clinical intervention technique in older adults.

Unfortunately, there are not enough studies by which you can extrapolate since the literature is insufficient. This is a limitation of our study.

- Discussion section of the paper (pages 8 and 9): One paragraph is needed discussing how these results should be seen in the light of the different types of dementia, e.g., Alzheimer dementia, dementia with Lewy bodies, frontotemporal dementias, vascular dementia, and mixed dementia? Are the cognitive reserve findings of the chess game the same for the different types of dementia or are they better in one specific kind of dementia?

Unfortunately, there are not enough studies by which you can extrapolate since the literature is insufficient. This is a limitation of our study.

- Discussion section of the paper (pages 8 and 9): The authors should discuss the methodological issue of “retrospective” versus “prospective” studies and why this is important in interpreting the results. Other methodological factors that could perfectly affect the study results are level of education, profession, culture, etc. I think these methodological affecting factors should be discussed more into detail. They are particularly important to correctly interpret the study results and are also very important with respect to future research.

Unfortunately, there are not enough studies by which you can extrapolate since the literature is insufficient. This is a limitation of our study.

- Discussion section of the paper (at the end, page 9, line 263): The authors should add a paragraph, discussing the limitations of the present study (see also my previous point 6).

Thanks for your appreciations. After reviewing this section we have made the following modifications: 

As part of the limitations of this Scoping Review,no studies that reflect the benefits of chess in the different dementia types (AD, dementia with Lewy bodies, frontotemporal dementia, vascular dementia and mixed dementia) were found. In addition, we did not found publications comparing the benefits of playing chess throughout life and playing chess just in older age.Finally, some of the studies we found did not define independent variables (sex, age, level of studies, profession, culture, etc.) of the dependent ones (playing chess and developing dementia).

- Discussion section of the paper (at the end, page 9, line 263): The authors should add a paragraph discussing what future research is needed based on the present study results, in order to make progress in the field. Please don’t forget to add the new references to your reference list.

- Conclusions (page 9, lines 279-284): Although I like your PhD project very much, this section should be deleted in the Conclusions section of the present paper.

Thanks for your appreciations. After reviewing this section we have made the following modifications: 

Considering our results, the practice of chess is a protective tool in the development of dementia from a preventive perspective and even though the evidences are weak to demonstrate its condition as a protective factor, other evidences indirectly related to Chess and based on its Cognitive Stimulating functions suggest it may work as a protective factor.

- Reference list (pages 10-12): In general, the authors did a good job with respect to the Reference list. However, several references need to be corrected according to the reference style of International Journal of Environmental Research and Public Health.

Thanks for your contributions. After reviewing this section we have made the following modifications:

References: 

1.       World Health Organization. Available online: http://www.who.int/mediacentre/factsheets/fs362/es/(accessed on 1 October 2016).

2.       Springer Nature. Disease Prevention-Nature. Available online: https://www.nature.com/subjects/disease-prevention(accessed on 26 February 2019).

3.       Medline. Early Diagnosis. MeSH–NCBI. Available online: https://www.ncbi.nlm.nih.gov/mesh/68042241(accessed on 26 February 2019).

4.       Medline. Protective Factor. MeSH-NCBI. Available online: https://www.ncbi.nlm.nih.gov/mesh/68042241(accessed on 26 February 2019).

5.      Coyle, J.T. Use it or lose it – do effortful mental activities protect against dementia? N Engl J Med. 2003, 348(25):2489-2490.

6.      Kienja, A., Frydecka, D., Adamowski, T., Bickel, H., Reynish, E., Prince, M. Epidemiological studies of cognitive impairment and dementia across Eastern and Middle European countries (epidemiology of dementia in Eastern and Middle European Countries). Int J Geriatr Psychiatry.2013, 26(2):111-7. 

7.      Stern, Y. Cognitive reserve in ageing Alzheimer’s disease. Lancet Neurol.2012, 11(11):1006-1012.

8.      Armstrong, R., Hall, B.J., Doyle, J. & Waters, E. Scoping the scope of a Cochrane review. J Public Health 2011, 33(1):147-55.  

9.       Colquhoun, H.L., Levac, D., O'Brien, K.K., Straus, S., Tricco, A.C., Perrier, L. et al. Scoping reviews: time for clarity in definition, methods, and reporting. J Clin Epidemiol. 2014, 67(12):1291-4.

10.   Scarmeas, N., Levy, G., Tang, M.X., Manly, J., Stern, Y.. Influence of leisure activity on the incidence of Alzheimer’s disease. J Neurology.2001, 57(12):2236-2242.

11.   Fabrigoule, C., Letteneur, L., Dartigues, J.F., Zarrouk, M., Commenges, D., Barber-Gateau, P. Social and leisure activities and risk of dementia: a prospective longitudinal study. J Am Geriatr Soc.1995, 43:485-490.

12.    Dowd, S.B., Davidhizar, R. Can mental and physical activities such as chess and gardening help in the prevention and treatment of Alzheimer's. J Pract Nurs.2003, 53:11-13.

13.    Derby, C.A., Katz, M.J., Lipton, R.B., Hall, C.B. Trends in dementia incidence in a birth cohort analysis of the Einstein Aging Study. JAMA Neurol. 2017.

14.    Wilson, R.S., Scherr, P.A., Schneider, J.A., Tang, Y., Bennet DA. Relation of cognitive activity to risk of developing Alzheimer disease. J Neurol. 2007, 69(19):11-1920.

15.    Spector, A., Woods, B., Orrell, M. Cognitive stimulation for the treatment of Alzheimer’s disease. Expert Rev Neurother. 2008, 8(5):751-757.

16.    Dartigues, J.F, Foubert-Samier, A., Le-Goff, M., Vittard, M., Amieva, H., Orgogozo, J.M. Playing board games, cognitive decline and dementia: a French population-based cohort study. BMJ Open. 2013, 3:1-7.

17.   Selkoe, D.J. Translating cell biology into therapeutic advances in Alzheimer's disease. J Nature. 1999, 399:23-31.

18.   Friedland, R.P., Fritsch, T., Smyth, K.A., Koss, E., Lerner, A.J., Chen, C.H. Patients with Alzheimer’s disease have reduced activities in midlife compared with healthy control-group members. Natl Acad of Sci. 2001, 98(6):3440-3445.

19.    Archer, H.A., Schott, J.M., Barnes, J., Fox, N.C., Holton, J.L., Revesz, T., Cipolotti, L. Knight’s move thinking? Mild cognitive impairment in a chess player. Neurocase.2006, 11(1):26-31.

20.   Li, K., Juang, J., Qiu, L., Yang, X., Huang, X., Lui, S. A multimodal MRI dataset of professional chess players. Sic Data. 2009, 1:2. 

21.    Lin, Q., Cao, Y., Gao, J. The impacts of a GO-game (Chinese chess) intervention on Alzheimer disease in a Northeast Chinese population. Front Aging Neurosci.2015, 7:1-10.

22.   Lee, B., Park, Y., Jung, W.H., Kim, H.S., Oh, J.S., Choi, C.H. White matter neuroplastic changes in long-term trained players of the game of “Baduk” (GO): A voxel-based diffusion-tensor imaging study. Neuroimage. 2010, 52:9-19.

23.   Gatz M. Educating the brain to avoid dementia: can mental exercise prevent Alzheimer’s disease? PLoS Med. 2005, 2:1. 

24.   Wilson, R.S., Mendes-de-Leon, C.F., Barnes, L.L., Schneider, J.A., Bienias, J.L., Evans, D.A. Participation in cognitively stimulation activities and risk of incident Alzheimer disease. JAMA. 2002, 287:742-748.

25.   Fabrigoule, C. Do leisure activities protect against Alzheimer’s disease? Neurology. 2002, 1: 11.

26.    Wang, H.X., Xu, W., Pei, J.J. Leisure activities, cognition and dementia. BBA. 2012, 1822(3):482-491.

27.   Blazer, D.G. Brain-stimulating habits linked to lower Alzheimer’s protein levels. Duke Med Health News. 2012, 18(4):3.

28.    Cimarra, M. Checkmating Alzheimer’s Disease. Available online: http://en.chessbase.com/post/checkmating-alzheimers-disease-210513(accessed on 1 October 2016). 

29.    Akbaraly, T.N., Portet, F., Fustinoni, S., Dartigues, J.F., Artero, S. Leisure activities and the risk of dementia in the elderly. Neurology. 2009, 73:854-861.

30.    Helzner, E., Scarmeas, N., Cosentino, S., Portet, F., Stern, Y. Leisure activity and cognitive decline in incident Alzheimer disease. Arch Neurol. 2007, 64(12):1749-1754.

31.   Fritsch, T., Smyth, K.A., Debanne, S.M., Petot, G.J., Friedland, R.P. J Geriatr Psychiatry Neurol. 2005, 18:134-141.

32.   Dowd, S.B. & Root, A. What chess has given us? Academic Erchange Quarter. 2003, 7, 1.

33.   Chiu, Y.C., Huang, C.Y., Kolanowski, A.M., Huang, H.L., Shyu, Y., Lee, S.H., Lin, C.R., Hsu, W.C. The effects of participation in leisure activities on neuropsychiatric symptoms of persons with cognitive impairment: A cross-sectional study. Int J Nurs Stud. 2013, 50:1314-1325.

34.   Gilbert, P., McEwan, K., Catarino, F., Baiao, R., Palmeira, L. Fears and happiness and compassion in relationship with depression, alexithymia and attachment security in a depressed sample. Br J Clin Psychol. 2014, 53:228-244.

35.   Li, L., Chang, H.J., Yeh, H., Hou, C.J.Y., Tsai, C.H., Tsai, J.P. Factors associated with leisure participation among the elderly living in longterm care facilities.Int J Gerontology. 2010, 4 (2), 69-74.

36.   Ball, Q., Berch, D.B., Helmers, K.F., Jobe, J.B., Leveck, M.D. Effects of cognitive training interventions with older adults: A randomize controlled trial. JAMA. 2002, 288:2271-2281.

37.   Stern, C., Munn, Z. Cognitive leisure activities and their role in preventing dementia: a systematic review. Int J Evid Based Health. 2010, 8:2-17.

38.   Verghese, J., Lipton, R.B., Katz, M.J., Hall, C.B., Derby, C.A., Kulansky, G. Leisure activities and the risk off dementia in the elderly. N Engl J Med. 2003, 348:2508-2516.

39.   Rajan, K.B., Wilson, R.S., Weuve, J., Barnes, L.L. & Evans, D.A. Cognitive impairment 18 years before clinical diagnosis of Alzheimer disease dementia. Neurology 2015, 85(10):898-904.

Minor revisions:

- Somehow there is no subheading 2.3? You use “2.2.- Literature search” on page 3 and “2.4.- Data extraction” on page 4.

Thanks for your correction. This section has already been modified.

- On page 9, line 274 à “recomended” should be “recommended”. Please correct.

Thanks for your correction. This section has already been modified

Round 2

Reviewer 1 Report

Recommendation:

- instead of using term "Dementia people" to use either "persons with dementia", either "patients with dementia syndrome"

Reviewer 2 Report

Dear Authors, Your paper has improved significantly. Therefore, I now recommend publication.